

# Stories told by corals, algae, and sea-urchins in a Mesoamerican coral reef: degradation trumps succession

Isael Victoria-Salazar[1,2], Edgar J. González[2], Jorge A. Meave[2], Miguel-Ángel Ruiz-Zárate[1] and Héctor A. Hernández-Arana[1]

[1] Departamento de Sistemática y Ecología Acuática, El Colegio de la Frontera Sur, Chetumal, Quintana Roo, Mexico
[2] Departamento de Ecología y Recursos Naturales, Facultad de Ciencias, Universidad Nacional Autónoma de México, Coyoacán, Mexico City, Mexico

## ABSTRACT

Understanding the mechanisms that allow the permanence of coral reefs and the constancy of their characteristics is necessary to alleviate the effects of chronic environmental changes. After a disturbance, healthy coral reefs display trajectories that allow regaining coral cover and the establishment of framework building corals. Through a comparative approach, in a patch reef partially affected by a ship grounding, we analyzed the successional trajectories in affected and unaffected sectors. Fleshy algae (which do not promote the recruitment of corals) dominated the reef surface irrespective of the impact of the ship grounding incident. *Acropora* species had near-zero contributions to community structure, whereas non-framework building corals like *Porites* sp. had a slightly higher recruitment. Cover of coral and calcareous crustose algae decreased over time, and neither the latter nor adult coral colonies had any effect on the occurrence probabilities of small corals. Sea urchin (*Diadema antillarum*) densities were generally low, and thus unlikely to contribute to reverting algal dominance. The successional trajectories of the community in the impacted and non-impacted sectors of the coral patch reef agree with the inhibition successional model, leading to the development of a degraded state dominated by fleshy algae. It is probable that the stability and resilience of this degraded state are high due to the ability of fleshy algae to monopolize space, along with low coral recovery potential.

## INTRODUCTION

In healthy reefs, post-disturbance recovery is featured by a steady increase in coral cover derived from lateral growth or from colonization of larvae from other sources, while fleshy algae cover is kept low by herbivores (*Duarte et al., 2015*; *Hartmann, Marhaver & Vermeij, 2018*; *Gouezo et al., 2020*). The reduction of competition for space with fleshy algae mediated by sea urchins (*e.g., Diadema antillarum* in the Caribbean) and herbivorous fish (*e.g.*, parrotfish) contributes to the maintenance of available substrate for coral recruitment (*Aronson & Precht, 2000*; *Edmunds & Carpenter, 2001*; *Do Hung Dang et al., 2020*). Also,

Corresponding author
Jorge A. Meave,
jorge.meave@ciencias.unam.mx

availability of colonizable substrate leads to the successful settlement, establishment, and survivorship of coral species (like *Acropora palmata*, *A. cervicornis* and *Orbicella* spp. in the case of Caribbean reefs), contributing to reef accretion (*Toth et al., 2019*). However, as all coral reefs of the world have been affected by human activities (*Riegl & Glynn, 2020*) they face increasingly harsher conditions due to the augmented frequency and intensity of disturbances (*Arnold & Steneck, 2011*; *Doropoulos et al., 2012*; *Webster et al., 2013*). These changes in disturbance attributes have resulted in a reduced capacity to recover coral-dominated states (*Hartmann et al., 2013*; *Gil et al., 2016*; *Bruno, Côté & Toth, 2019*). In the Caribbean region, increases of fleshy algae cover have been a constant phenomenon (though dominance by other groups is also possible; *Norström et al., 2009*), ultimately reducing the chances for corals to find suitable surface for settlement, survival, and growth (*Pineda & Caswell, 1997*; *Ferrari, Gonzalez-Rivero & Mumby, 2012*; *Doropoulos et al., 2016a*; *Evensen et al., 2019*).

Ship groundings caused by accident or negligence are complex disturbances with profound effects on coral reef structure (*Precht, Aronson & Swanson, 2001*). The complexity of a ship grounding depends mostly on ship attributes (size, weight, speed), features of the coral reef affected (depth, size, and physical structure, but also its health), and the environmental conditions at the moment of impact (tides, waves, or storms) (*Jaap, 2000*; *Nguyen et al., 2011*). Among the impacts of ship groundings on coral reefs, physical damage and pollution by oil or cargo spills are noteworthy (*Precht, Aronson & Swanson, 2001*; *Schroeder et al., 2008*; *Lirman et al., 2010*), with contrasting effects on the system's dynamics. In addition to regrowth and fragmentation, recovery can also occur through intense coral recruitment in as little time as five years (*Gittings, Bright & Holland, 1990*) or be completely absent in the same timeframe (*Lirman et al., 2010*). Overall, following ship groundings two general complementary patterns have been reported: (1) the longer the time after the disturbance, the larger is coral cover (*Riegl, 2001*); and (2) opportunistic species start pre-empting the available space (*Smith, 1985*). In a context of widespread coral reef degradation, a ship-grounding, when having purely physical effects (*i.e.,* with chemical damage being absent), could open new opportunities to shift the dynamics of a degraded coral reef towards one favoring a state dominated by corals, especially those that increase the reef-framework accretion. For the latter to occur, an optimal cover of coral recruitment surface is required, which is provided by crustose calcareous algae (CCA) (*Ritson-Williams et al., 2010*; *Ritson-Williams et al., 2014*), along with the recruitment of reef-building corals, and an efficient control of fleshy algae.

Here, we study a coral reef patch that was designed as a no-take area after it was partially affected by a ship-grounding in 1997. We ask whether in the reef's affected portion a community dominated by corals is developing, especially of coral species capable of contributing to reef accretion or, on the contrary, the developing community continues to be representative of degraded reefs. To answer this question, we established two objectives. First, we assessed the occurrence of small colonies as a proxy of coral recruitment (*Edmunds, 2021*), particularly of framework building corals, both in the impacted and non-impacted sectors of the patch reef, relative to the cover of adult coral colonies and different algal groups (CCA, fleshy algae). Second, we assessed the relationship of algae that either promote
(*i.e.*, CCA) or prevent (*i.e.*, fleshy, turf; *Birrell, McCook & Willis, 2005*; *Arnold & Steneck, 2011*; but see *Mumby et al., 2013*), branching coralline and crustose non-coralline algae, hereinafter referred to collectively as non-CCA) with juvenile coral occurrence. Derived from the current degradation patterns in Caribbean coral reefs (*Alvarez-Filip et al., 2015*), we expected to find evidence for a relatively unsuccessful re-establishment of a healthy coral-dominated patch reef.

## MATERIALS & METHODS

### Study site

This study was conducted in the 'Parque Nacional Costa Occidental de Isla Mujeres, Punta Cancún y Punta Nizuc' (hereinafter Cancún National Park or CNP), in northern Quintana Roo state, Mexico (21°09′41.5″N, 86°44′27.6″W) (Fig. 1A). In December 1997, the *Leeward* vessel (159 m long, 25,600 tons, capacity for 650 passengers; *Wikipedia, 2022*) impacted the Cuevones patch reef (Fig. 1B), dislodging the biota of 465.5 m$^2$ from its substrate. The vessel's integrity was not compromised (neither oil, fuel nor cargo spilled into the water), thus non-physical long-term impacts were discarded. Prior to the ship grounding, the dominant species in the patch were *Porites astreoides*, *Acropora palmata*, *A. cervicornis* and *Orbicella annularis* (*Reef Keeper International, 1997*). Yet, just before the impact, the patch had begun to show early signs of its transit towards an algae-dominated state (*Victoria-Salazar et al., 2017*).

As only one portion of the patch was affected and the impacted sector was a long but relatively narrow strip, a contiguous non-impacted sector along its longest axis was used as a reference and assumed to be the closest source of larvae for the impacted sector (Fig. 1C). Every six months (February and August 2013, and February and September 2014), 20 out of 40 1-m$^2$ permanent quadrats, marked with stainless steel pins, were randomly selected within each sector and used to record the cover of juvenile corals, adult corals, and algae, as well as the abundance of the sea urchin *Diadema antillarum*. The selection of the non-impacted reference sector relied on three grounds. First, the analysis of historical records showed that in 1996, a year before the ship-grounding, corals and algae occupied ca. 40% of the surface, each (*Victoria-Salazar et al., 2017*). In August 1997, still before the impact, coral cover had already decreased to ca. 25% while algae cover had increased to ca. 50%. Two months after the impact, coral cover had increased to ca. 30% in both the impacted and unimpacted sectors. From February 1997, coral cover decreased to less than 10% of the surface, while algae cover had increased to ca. 85% by 2007. The same study demonstrated changes in the biotic composition of this reef, mostly due to the loss of *Orbicella annularis* (*Victoria-Salazar et al., 2017*). Secondly, passive transport of coral propagules (coral larvae or fragments) is strongly linked to marine currents (*Pearson, 1981*; *Pineda, Hare & Sponaugle, 2007*); in Cuevones, the direction of the dominant current facilitates propagule transport from the non-impacted to the impacted sector (Fig. 1C). Thirdly, colonization often starts from the margins between impacted and non-impacted areas (*Fox & Caldwell, 2006*). As only one portion of Cuevones was affected, the non-impacted sector is the portion of the patch reef having the longest contact with
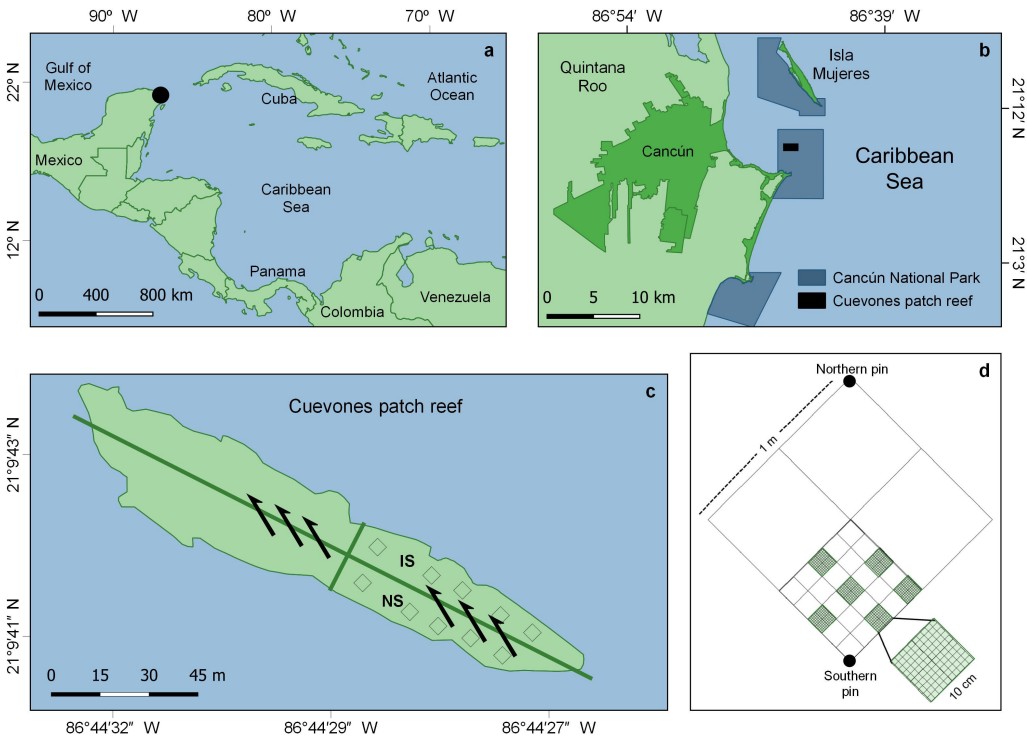

**Figure 1 Study area and sampling details.** (A) General and (B) detailed location of the Cancún National Park (CNP), Quintana Roo, eastern Mexico. (C) Cuevones patch within CNP and location of the impacted (IS) and the non-impacted (NS) sector, where the study was conducted; the arrows show the direction of the current. (D) Example of a sampling unit for coral and algal cover, and sea urchin density. Modified from *Victoria-Salazar et al. (2017)*.

the impacted sector; thus, the non-impacted sector is arguably the main (if not the only) local source of propagules.

## Data collection
### Adult coral cover and sea urchin abundance

Randomly selected 1-m² quadrats were delimited on the reef surface and information for all complete or fractional adult colonies occurring in them was recorded. For each colony two diameters (the longest and one perpendicular to it) were measured with a graduated tape. Colony cover area calculations were based on these diameters (rounded to the nearest cm) by approximation to an ellipse (Eq. (1)):

$$Colony\ cover = ((dmax \times dper)/4) \times \pi \qquad (1)$$

where *dmax* is the maximum diameter and *dper* is the diameter perpendicular to it. Next, proportional coral cover was obtained by dividing coral area (cm²) by the quadrat's area (10,000 cm²). All *Diadema antillarum* individuals roaming within each quadrat were tallied.

### Juvenile corals

To search for juvenile corals, a 50 × 50 cm PVC frame was attached to the southern pin of the twenty 1-m² quadrats placed in each sector (Fig. 1D). The PVC frame was divided into twenty-five 10 × 10 cm units. In each quadrat we carefully searched for all hard-coral juvenile colonies with a diameter ≤ 4 cm (*Acosta, Dueñas & Pizarro, 2011*; *Edmunds, 2021*). While we recognize that colony fragmentation, mortality or contraction may obscure juvenile coral counts, we also based our decision to consider a colony depending on its context, *i.e.,* we examined whether each juvenile coral was associated with a colony fragment, or with colonies with signs of tissue mortality. Once a juvenile coral was sighted it was identified to genus level. Colony cover was approximated to a circle and a juvenile coral proportional cover (*scp*) was calculated for each genus, as follows:

$$scp = \Sigma_i csc_i / sampling\ area \tag{2}$$

where $csc_i$ is the cover (cm²) of the *i*-th juvenile coral and sampling area is 2,500 cm².

### Algal cover

We assessed algal cover in five randomly selected 10 × 10 cm sampling units, in which we recorded the cover of the benthic groups underneath an acrylic plate (to the nearest cm²) with a 1-cm resolution grid. Algal cover was classified into two categories according to their functional group: (1) algae promoting coral recruitment, which only included CCA; and (2) non-CCA algae, which included the aforementioned fleshy macroalgae (FA), turf algae (TA), branched coralline algae (BCA), and crustose non-coralline algae (CnCA). We used these data to independently calculate the proportional contribution of each algal category (*pac*) to reef surface by quadrat in the space not occupied by adults or recruit coral colonies:

$$pac_{CCA} = (1 - crp)\Sigma_i pac_i^{CCA}/500, \text{and}$$

$$pac_{non-CCA} = (1 - crp)\Sigma_i(pac_i^{FA} + pac_i^{TA} + pac_i^{BCA} + pac_i^{CnCA})/500 \tag{3}$$

where $pac_{CCA}$ is the proportional cover of CCA, $pac_{non-CCA}$ is the proportional cover of non-CCA algae, $ac_i^j$ is the cover of algal functional group *j* (*i.e.,* CCA, FA, TA, BCA or CnCA) in the *i*-th sampling unit, *crp* is the proportion occupied by corals, and 1 - *crp* is the proportion of space not occupied by corals.

## Data analysis
### Benthic community structure

Through a multivariate approach, we assessed the current differences and similarities in the cover of the benthic community (which includes hard corals and algal groups) developing in the non-impacted sector and the impacted sector 16 years after the ship grounding. Using a dissimilarity matrix based on the Bray–Curtis index, a two-way PERMANOVA test was computed (9,999 permutations), with time and condition as fixed factors. When significant differences were detected, pairwise comparisons were performed. All tests were computed with PRIMER V6 and the PERMANOVA+ add-on (*Clarke & Gorley, 2006*). We used a

principal coordinate analysis (PCoA) for time and condition (impacted *vs.* non-impacted) to achieve a clear visualization of between-sector differences; we also plotted benthic groups strongly correlated with the ordination axes (Spearman $r_s > 0.5$).

### Juvenile corals

To assess the occurrence probabilities of juvenile framework building corals (like *Acropora* or *Orbicella* species) and non-framework building corals (like *Porites* or *Agaricia* species) and their relationship with sector condition (impacted and non-impacted), CCA and adult coral covers, we tested 12 models (Table S1) using binomial (logit-link) regressions with the 'stats' package in R (*R Core Team, 2017*). The selection of the best model was based on the Bayesian Information Criterion (BIC); if more than one model were similarly supported, a full averaging approach was used based on BIC weights (*Symonds & Moussalli, 2011*).

### Changes over time

The cover proportions of the surveyed benthic groups were classified into four categories nested in pairs: (1) proportion of coral cover, irrespective of its role in reef development; (2) proportion of framework building coral cover (nested to category 1); (3) proportion of algae, irrespective of categorization (*i.e.,* CCA or non-CCA); and (4) proportion of CCA (nested to category 3). All proportional data were transformed to fit the standard interval unit (using Eq. (4)) (*Cribari-Neto & Zeileis, 2010*), except for algae cover proportion, which fitted the interval in their original form:

$$p' = (p(n-1)+0.5)/n \tag{4}$$

where $p$ is the observed proportion, and $n$ is sample size.

The proportion of each category was modelled using a mixed beta (logit-link) regression, with time, sector condition, and sea urchin abundance as fixed factors, and quadrat as random intercept (*Cribari-Neto & Zeileis, 2010*). To fit the model to the data, a Markov Chain Monte Carlo (MCMC) method was performed using JAGS (*Plummer, 2003*) *via* the 'R2jags' package (*Su & Yajima, 2021*) in R. For each posterior distribution 60 chains were computed, each with 50,000 iterations (with a burn-in of 10,000 iterations and a thinning of 10). Diffuse normal priors were used for the fixed and random parameters. In turn, for the standard deviation a half-Cauchy distribution was used. In total, eight models were fitted for each response variable (Table S2); in this case, the null model considered time and condition.

## RESULTS

### Benthic community structure

The community structure differed significantly between the impacted and non-impacted sectors of the patch reef (pseudo-$F = 6.59$, d.f. $= 1$, $p < 0.001$), and changed over time (pseudo-$F = 3.60$, d.f. $= 3$, $p < 0.001$). Posterior pair-wise comparison of sampling dates revealed significant differences among them (most $P$-values were $< 0.01$) except between February and August 2013. The groups that had correlations $> 0.5$ with PCoA axes were *Porites astreoides*, *Millepora complanata* and turf algae, all of them with increasing covers

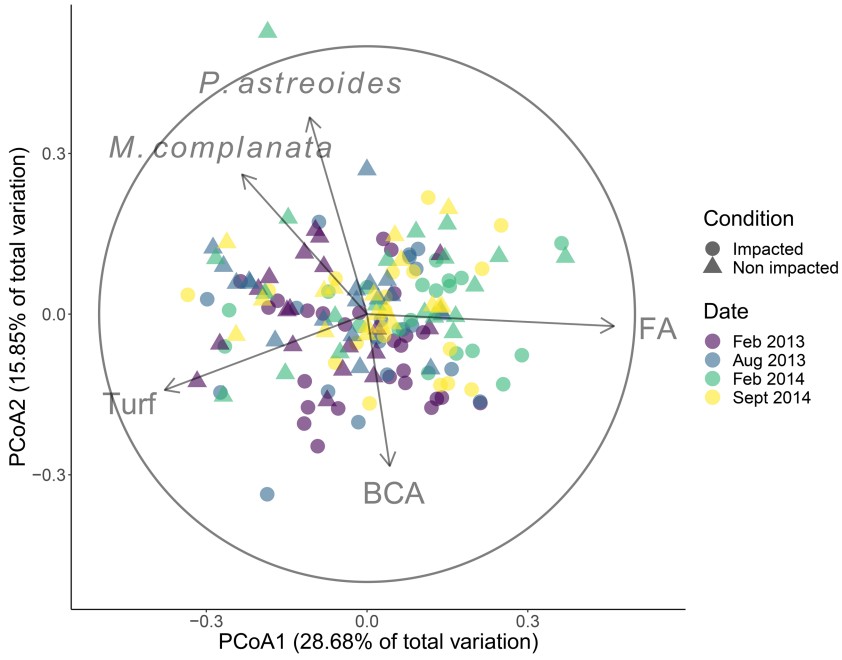

**Figure 2** **Results of the principal coordinates analysis (PCoA).** Community cover recorded in four dates in the impacted and non-impacted sectors of the Cuevones patch reef. Lines represent Spearman correlations of those benthic groups with $r_s > 0.6$. BCA, branched coralline algae; FA, fleshy algae. Full species names: *Millepora complanata* and *Porites astreoides*.

towards non-impacted sector samples, while fleshy algae and branched calcareous algae were positively related with the impacted sector (Fig. 2). Over the entire duration of the study, fleshy algae dominated in both sectors, followed by turf algae (Figs. S1 and S2). Regarding corals, *Porites astreoides* was the most important contributor to the benthic cover of all recorded species. The contribution of framework building corals to benthic cover was negligible in both sectors.

### Juvenile corals

Over the two-year period, only 79 small (juvenile) corals were recorded. *Acropora* (mean diameter $\pm$ SD = 2.48 $\pm$ 0.715 cm, density in the impacted sector (IS) = 4 ind 20 m$^{-2}$, density in the non-impacted sector (NS) = 2 ind 20 m$^{-2}$) was the only framework building genus occurring in both sectors, and its contribution to recruitment was minimal (Fig. S3). Conversely, *Porites* (mean diameter $\pm$ SD = 2.35 $\pm$ 0.978 cm, $n$ = 61, density in IS = 35 ind 20 m$^{-2}$, density in NS = 26 ind 20 m$^{-2}$) was the dominant genus in the small coral community in both sectors at all sampling dates. Other genera recorded were *Agaricia* (mean diameter $\pm$ SD = 2.60 $\pm$ 0.710 cm, density in NS = 8 ind 20 m$^{-2}$) and *Millepora* (mean diameter $\pm$ SD = 2.05 $\pm$ 0.764 cm, density in IS = 2 ind 20 m$^{-2}$, density in NS = 2 ind 20 m$^{-2}$). Models showed that framework building corals had the lowest occurrence probabilities on the reef and a lack of relationships with sector condition, adult colony cover, or the cover CCA (Fig. 3A, Table S3). Our models also showed higher occurrence
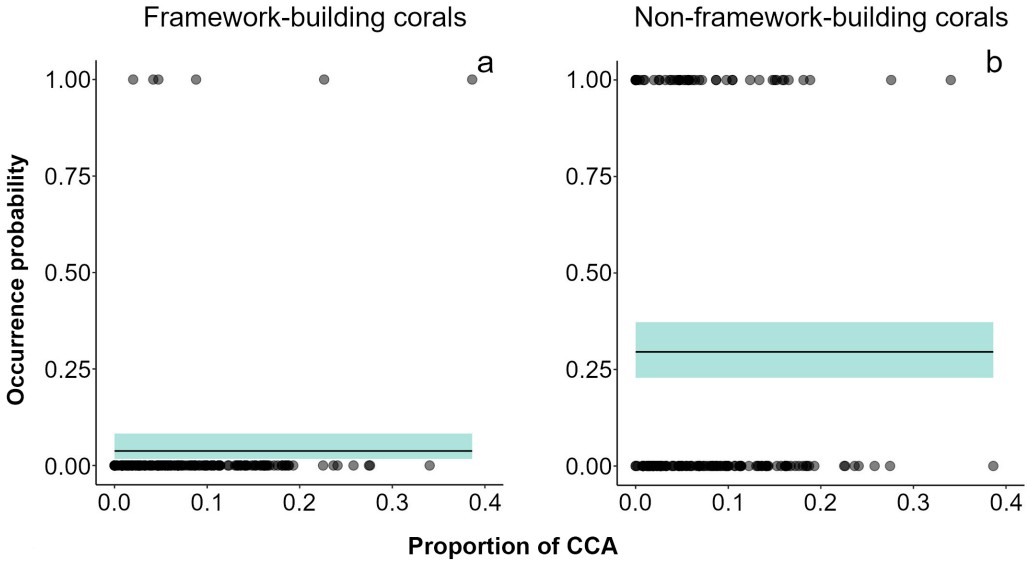

**Figure 3** **Recruit occurrence probabilities.** (A) Framework building and (B) non-framework building corals. The average did not show any relationship between recruitment surface and occurrence probability of recruits. Shaded areas represent 95% confidence envelopes.

probabilities for non-framework building than for framework building corals; however, no single explanatory variable had an effect on this probability (Fig. 3B, Table S3).

## Short-term changes of functional groups

Algae were the main contributors to cover (ca. 70%) in the four sampling dates (Figs. 4E–4H). Non-CCA algae were the main contributors to algal cover with > 50% of the area in both sectors at all sampling dates. By contrast, CCA did not exceed 15% of proportional cover in both sectors at the four sampling dates (Figs. 4E, 4F). Mean coral cover ranged between 10 and 16% in the impacted sector, and between 20 and 26% in the non-impacted one. Framework building corals had the lowest proportional cover during the entire study period (IS, 1.3–3.4%; NS, 5–10%; Figs. 4A, 4B), whilst non-framework building corals had higher covers (IS, 10–14%; NS, 14–21%; Figs. 4C, 4D). Across sampling dates, mean *Diadema antillarum* density ranged between 0.04 and 0.81 ind m$^{-2}$, with virtually no difference between sectors.

Models for all benthic groups achieved a good mixing of the 60 MCMC chains, and model validation did not reveal residual dispersion problems. BIC weights showed that M1 (*i.e.,* sector condition and time) had the highest value for three of the four variables (Table S4), and it represented the only candidate model for coral proportional cover ($w = 0.891$, Table S4). For framework building corals, algae as a whole and CCA alone, the full average model showed that condition and time were the strongest covariates explaining changes in all benthic categories examined (Table S5). Models showed that framework building corals and non-CCA had the lowest and the highest cover proportions, respectively, among all response variables and throughout the study period (Figs. 4A, 4B, 4G, 4H). The model for framework building corals showed that their cover proportion decreased in both sectors;

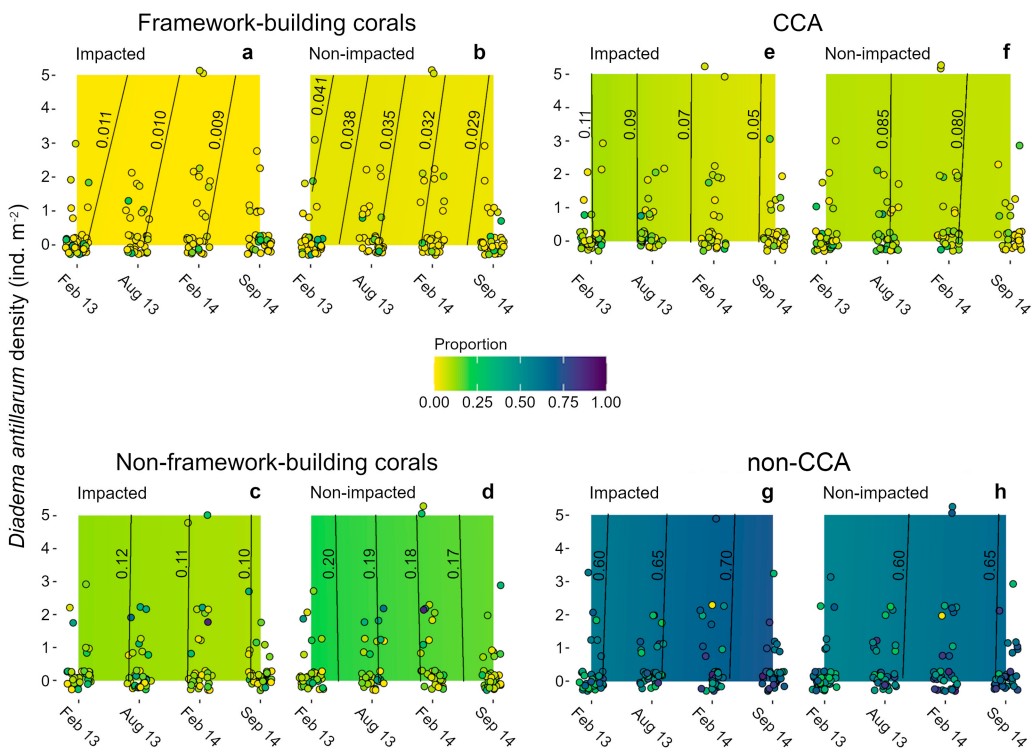

**Figure 4 Heat maps of estimated successional trend of cover of four functional groups.** (A, B) framework- and (C, D) non-framework building corals, and for (E, F) CCA and (G, H) non-CCA as functions of three explanatory variables: time (the four sampling dates), sea urchin densities, and sector condition (impacted *vs*. non-impacted). Dots represent the original cover proportions recorded in the field. Lines in the heat maps are isolines of cover proportion (indicated by the numbers next to them) and their position denotes sea urchin density-time interactions; isolines leaning to the right (*i.e.,* positive slope) denote a positive interaction, lines leaning to the left (*i.e.,* negative slope) denote a negative interaction, and fully vertical lines denote no interaction between these two predictor variables. Note that most samples had a sea urchin density of 0 ind m$^{-2}$. Some jittering was added to recorded data points to minimize overlap.

in the impacted sector there was a 0.2% reduction, whilst in the non-impacted sector cover reduction was considerably higher (1.2%) (Figs. 4A, 4B); in the case of non-framework building corals, cover reduction was similar for both the impacted and non-impacted sector (2 and 3%, respectively) (Figs. 4C, 4D). Regarding algal categories, non-CCA increased their cover proportion over time and were larger in the impacted sector (Figs. 4G, 4H). By contrast, the cover proportion of CCA decreased over time (Figs. 4E, 4F). The effects of *D. antillarum* on all four response variables assessed were modest, probably due to the overall low abundances within quadrats, reflected in a high frequency of quadrats devoid of sea urchins compared with a few quadrats with sea urchin densities as high as 5 ind m$^{-2}$ (Fig. 4, Table S5). Interactions between time and sea urchin densities were variable but overall weak for all response variables, except in the case of cover proportion of framework building corals, as shown by the inclined isolines in Figs. 4A, 4B; positive slopes in these isolines imply that higher sea urchin densities delayed the decrease of coral cover over time

by about two months. For example, the mean cover proportion recorded in February 2013 (1.1%) was reached between April and May in quadrats with a sea urchin density of 5 ind m$^{-2}$ (Fig. 4A).

## DISCUSSION

Based on the information gathered in a patch reef 16 years after it was affected by a ship grounding, our results confirmed that this community is still dominated by fleshy algae and non-framework building corals, as anticipated. This implies that the sector affected by the incident, despite the no-take protection status it enjoys since, is dominated by species that do not contribute to the recovery of a physically complex state. Instead, coral species with low contribution to reef accretion, like *Porites astreoides,* and non-CCA, are the main contributors to live cover in this sector. The present condition is characterized by a low recruitment probability (as inferred from the low juvenile coral densities), low cover, and the continued loss of framework building corals, all of which is reflected in the increasing cover of non-framework building corals but more importantly, in the dominance of non-CCA. Moreover, our results showed that cover of non-CCA increased and gradually replaced non-framework building corals over the study period. Notably, this pattern was evident in both sectors regardless of disturbance history, the protection status of this marine area, or the presence of non-framework building coral species, such as *Porites astreoides.* Importantly, this process seems to be occurring also on a fraction of Cuevones currently under a coral restoration program (*Padilla-Sousa et al., 2018*), where coral cover has increased because of the introduction of coral fragments (*Perera-Valderrama et al., 2016*). Under these circumstances, Cuevones may not be able to attain a coral-dominated state without intervention. If regional factors causing detrimental changes (*e.g.*, coastal development or highly intense tourism) in coral reefs persist for a long time or become permanent, this and other Caribbean reefs will very likely continue to display algae-dominated community states, thus jeopardizing the success of coral restoration programs (*Ware et al., 2020*). Actually, there is evidence of steady reduction in coral cover and a concomitant increase in algal cover in the Caribbean region that seems to have started shortly after a massive mortality of *Diadema antillarum* was observed in the early 1980's. At that time, mean hard coral cover was 30% but by 2015 had decreased to ca. 12%, while algal cover increased from 17% to 25% or more during the same period (*Contreras-Silva et al., 2020*). The findings in Cuevones depicted in Fig. 4 are confirmatory of these trends in the Caribbean, with non-CCA cover being the highest recorded so far in this region, with values as high as 60% of the surface.

In Caribbean reefs, the loss of resilience and the consequent development of degraded states have been driven by a combination of anthropogenic stresses, ranging from eutrophication to decimation of herbivores, all of which has resulted in significant declines in framework building coral species populations (*Jackson et al., 2014*). This loss of coral cover releases space that becomes available to algal colonization, while negative feedback from herbivores remains low. In turn, the presence of non-CCA (*i.e.,* fleshy and turf algae) deters coral recruitment and reduces coral recruit survival (*Steneck, Arnold & Mumby, 2014*;
*Olsen, Sneed & Paul, 2016*). This algal group favors the dominance of disturbance-tolerant coral species with large ecological plasticity (*Alvarez-Filip et al., 2013*). Such a role has serious ecological implications for reef accretion and the ecosystem services provided by coral reefs (*Kennedy et al., 2013*; *Perry et al., 2013*; *Perry et al., 2015*). A further consequence of this replacement is the local extinction of corals, ultimately resulting in the absence of refuges necessary to sustain other reef inhabitants (*Vermeij, 2006*; *Green, Edmunds & Carpenter, 2008*; *Alvarez-Filip et al., 2013*; *Perry et al., 2015*).

The ultimate effect of the ship grounding on the Cuevones patch reef was to catalyze the shift in dominance in the impacted sector from a framework building coral-dominated state towards a degraded state dominated by non-CCA and non-framework building coral species (*Victoria-Salazar et al., 2017*). In this state, non-framework building species such as *Porites astreoides* would be able to colonize the reef surface due to its ability to develop healthy populations in harsh environments (*Green, Edmunds & Carpenter, 2008*; *Olsen, Sneed & Paul, 2016*). In this study, the analysis of occurrence probabilities of juvenile corals suggests that recruitment of framework building species (*i.e., Acropora* and *Orbicella* species) is so low that they make a near-zero contribution to coral community structure, and that those few corals that reach an adequate substrate did not thrive even under the infrequent circumstances where CCA cover was as high as 40% (Fig. 3A). Instead, non-framework building species had higher chances to contribute to the recruit community. Low recruitment of juvenile individuals of *Acropora* and *Orbicella* species is a common condition in Caribbean reefs, even in healthy reefs (*Rylaarsdam, 1983*); this situation becomes critical at present, given the lack of adult colonies of these species, which in earlier times were probably able to add new colonies to the reef through fragmentation (*Roff, 2020*).

Several factors have been recognized to drive coral larval settlement, among which sedimentation, interspecific competition with fleshy algae and other groups, available space for settlement, chemical cues, and connectivity with other reef communities that supply coral propagules are noteworthy (*Ferrari, Gonzalez-Rivero & Mumby, 2012*; *Tebben et al., 2015*; *Suchley & Alvarez-Filip, 2017*). At Cuevones, low recruitment results from a combination of several factors. First, autogenic recruitment could be limited due to the low proportional cover of adult colonies in both sectors (Fig. S1), which is reflected by the null contribution of adult coral cover as an explanatory variable for settlement success, irrespective of their contribution to reef accretion. Second, connectivity (as measured through genetic diversity among framework building coral populations), an important factor promoting coral reef resilience (*Elmhirst, Connolly & Hughes, 2009*; *Manikandan et al., 2017*), appears to be low (*Domínguez-Maldonado et al., 2022*), although the genetic evidence for such lack of connectivity is scarce. Causes of connectivity failure may be related to the reduction of larval sources in the Mesoamerican Reef System, where only one site (Limones reef, also in northern Quintana Roo) has been reported to sustain a healthy population of *Acropora palmata* (*Rodríguez-Martínez et al., 2014*), with a high potential fecundity but low recruitment (*Acevedo-Rosas, 2017*). A third factor apparently involved is related to the low amount of recruitment substrate available for framework building species, which also decreased over time (Figs. 4E, 4F). This latter factor could have an
enhanced effect in Cuevones; despite the abundance of non-framework building species and their high reproductive rates, the available space for coral larval settlement is dwindling and turning into a non-recruitment substrate (Figs. 4E–4H).

Settlement, survivorship, and growth opportunities for framework building coral recruits, which are more selective in choosing a settlement substrate than non-framework building species (*Olsen, Sneed & Paul, 2016*), strongly depend on the grazer-mediated reduction of algal cover (reviewed in *Adam et al. (2015)*). In this study, only a few quadrats had urchin densities as high as 5 ind m$^{-2}$, whereas quadrats with low abundances of this sea urchin were frequent, with densities ranging from 0 to 3 ind m$^{-2}$ (see Fig. 4). Sea urchin distribution depends on the complexity of the reef surface, with physically more complex reefs offering more refuge space (*Dunn et al., 2017*). Thus, given the dominance of sub-massive coral species observed at Cuevones, the only available space usable by sea urchins in the non-impacted sector are the crevices and the few remaining standing *Acropora palmata* skeletons. The low refuge availability leads to the exclusion of these herbivores, with negative impacts on coral recruitment (*Ruiz-Zárate & Arias-González, 2004*; *Doropoulos et al., 2016b*).

From a mechanistic perspective, community recovery depends on ecological succession and in coral reefs this process may take place in one of two ways. On the one hand, facilitation (*sensu Connell & Slatyer, 1977*) occurs when either CCA or facilitative biofilms establish right after the ship grounding (*Arnold & Steneck, 2011*), dominating the reef surface. Alternatively, inhibition mechanisms come into play when earlier colonizers prevent the settlement of coral larvae, which represent the late successional dominant species. The answer to the question of which mechanism of ecological succession operates in each coral reef largely depends on the identity of earlier algal colonizers (*Mumby et al., 2007*; *Doropoulos et al., 2016b*). In our study, proportional cover of non-CCA was up to eight-fold that of CCA in both sectors, and the models showed that CCA decreased over time. Hence, the Cuevones seascape seems unsuitable for corals to successfully settle on to. The type of benthic early successional groups is strongly coupled with herbivore densities (*Steneck, Arnold & Mumby, 2014*; *Doropoulos et al., 2016a*; *Mumby et al., 2016*). Our models also showed that, for the Cuevones patch reef, ecological succession in both sectors is driven by a stabilizing mechanism that promotes fleshy-algae dominance. Despite the no-take protection scheme offered by the marine protected area and the restoration efforts, modifying this situation in the current ecological context seems unlikely. Densities of the herbivore *D. antillarum* in both sectors over the entire study period were so low that its effect is very likely negligible, which explains the continuous increase in the proportion cover of non-CCA. In the Caribbean, such low densities, even under the protected condition of an MPA, have been linked to predation upon the sea urchin by fishes (*Harborne et al., 2009*) and a low inter-population connectivity, which together translate into low local larvae availability (*Carpenter & Edmunds, 2006*).

In summary, this study shows that the community that is developing in the damaged sector of the Cuevones patch reef has been following a successional trajectory that seems to be in accordance with a stabilizing mechanism (*Pulsford, Lindenmayer & Driscoll, 2016*), characterized by the dominance of coral recruitment inhibition processes, and conforms to

the inhibition successional model proposed by *Connell & Slatyer (1977)*. Cover of non-CCA has increased over time, occupying most of the substrate and reducing the availability of suitable space for coral establishment. By contrast, CCA cover, which is necessary for the successful establishment of framework building coral larvae, decreased over time, reducing even more the low natural rates of recruitment. It is important to note that these algal patterns are consistent across the different sectors in Cuevones, regardless of disturbance history. Previous studies have shown the existence of herbivore-mediated shifts between inhibition and facilitation mechanisms (*Doropoulos et al., 2016b*). At Cuevones, although higher *D. antillarum* densities were associated with reductions in non-CCA, the overall abundance of this herbivore was very low. Interestingly, however, quadrats having between one and three sea urchins showed a high variation in algal cover, suggesting that the abundance of this herbivore is currently below the minimum threshold needed for this species to have a regulatory effect on such cover (*Edmunds & Carpenter, 2001*; *Mercado-Molina et al., 2015*). The permanent analysis of ecological succession in this and other coral reefs affected by a variety of disturbance agents will provide insights into the processes leading to the loss of resilience in these complex and highly diverse ecological systems.

## ACKNOWLEDGEMENTS

IVS and HAHA thank the Cancún National Park staff for assisting with data collection in the field. Haydée López, Elizabeth Arista, Fernando Valdez, and Alejandro Vega assisted with data collection. Jorge Omar López kindly provided access to computing facilities.

### Funding

The Graduate Program in Biological Sciences (UNAM) and CONACyT (grant 229054) supported Isael Victoria-Salazar. CONABIO funded this study (grant JA009 to Héctor A. Hernández-Arana). Cancún National Park supported field-work and El Colegio de la Frontera Sur provided additional funding for field work. The funders had no role in study design, data collection and analysis, decision to publish, or preparation of the manuscript.

### Grant Disclosures

The following grant information was disclosed by the authors:
Graduate Program in Biological Sciences.
CONACyT: 229054.
Isael Victoria-Salazar.
CONABIO: JA009.
El Colegio de la Frontera Sur.

### Competing Interests

The authors declare there are no competing interests.

## Author Contributions

- Isael Victoria-Salazar conceived and designed the experiments, performed the experiments, analyzed the data, prepared figures and/or tables, authored or reviewed drafts of the article, and approved the final draft.
- Edgar J. González analyzed the data, prepared figures and/or tables, authored or reviewed drafts of the article, and approved the final draft.
- Jorge A. Meave conceived and designed the experiments, analyzed the data, prepared figures and/or tables, authored or reviewed drafts of the article, and approved the final draft.
- Miguel-Ángel Ruiz-Zárate conceived and designed the experiments, performed the experiments, analyzed the data, prepared figures and/or tables, authored or reviewed drafts of the article, and approved the final draft.
- Héctor A. Hernández-Arana conceived and designed the experiments, performed the experiments, analyzed the data, prepared figures and/or tables, authored or reviewed drafts of the article, and approved the final draft.

## Data Availability

The raw data is available in the Supplementary File.

## Supplemental Information

Supplemental information for this article can be found online at http://dx.doi.org/10.7717/peerj.14680#supplemental-information.

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
