# Peer review of "Stories told by corals, algae, and sea-urchins in a Mesoamerican coral reef: degradation trumps succession"

_PeerJ, doi:10.7717/peerj.14680_

## Round 0.1 · original submission · Major Revisions

First of all, I would like to apologize for the delay in the handling of this manuscript. Finding reviewers has taken significantly longer than it should have, and I thank you for your patience. The manuscript has now been assessed by two experts in the field who have taken the time to provide extensive and constructive feedback.

Overall, I agree with their assessment that this study is a well-written and great example of reef community recovery (or more specifically the lack thereof) for the region. While the reviewers identify come concerns with the presentation and interpretation of the study, they also make excellent suggestions as to how to overcome these limitations. I am confident that paying close attention to their feedback will greatly improve the clarity and narrative of the manuscript.

In addition to the reviewers’ comments, I would like to add one question regarding equation 1 used to calculate the colony cover in the manuscript. In its current form (Colony cover = (dmax × dper)/4) the equation is not that of an ellipse as stated in L131 and would greatly underestimate coral cover in the investigated areas. Could you please clarify or adjust the calculation as needed?

Reviewer 1 ·

Basic reporting

Review Victoria - Salazar et al

This is an interesting study of coral reef recovery following a ship strike. Such events are unfortunate opportunities to examine post-disturbance recovery resilience of coral reefs.

The evidence is clear that following a ship strike, colonization by macroalgae reduces the recruitment of corals. The demonstrably long duration of that recruitment inhibition effectively becomes an alternate stable state for that portion of the coral reef. The authors are correct in identifying this as a good example of the "Inhibition model" of succession defined by Connell and Slatyer 1977. However, there were other conclusions not supported by their data or the published literature. I will discuss those points below.

The authors assert that the shift to macroalgae following the ship strike resulted in a shift in coral recruitment from framework corals such as Acropora and Orbicella genera to species such as Porites astreoides which contributes less to coral reef frameworks. However, the authors seem to overlook the fact that Porities astreoides is often the most common coral recruit regardless of macroalgal abundance. Framework corals such as Acropora and Orbicella species rarely recruit regardless of the abundance of algae, Diadema, CCA or adult coral cover. This was well illustrated for Jamaica in the 1970s when Acropora and Orbicella species were abundant, Diadema populations were extremely high, coralline algae were abundant but macroalgae were rare (see Rylaarsdam 1983). In the Rylaarsdam study Agaricia, Leptoseris and Porities astreoides were the dominant juvenile coral and Acropora and Orbicella coral recruits were rare. Other recent studies of coral recruitment in algal free, CCA rich reefs in the Caribbean such as Bonaire determined P. astreoides and Agaricia were the dominant recruiting corals and few if any framework species of Acropora or Orbicella were recorded.

Important to clarify that fleshy algae inhibit the recruitment of all corals not just the "framework-building corals" as asserted in the abstract. The assertion (line 331) that "settlement, survivorship and growth" of Acropora palmata and Orbicella annularis "depend strongly on grazer-mediated reduction of algal cover" lacks support. The cited Doropoulos et al 2016b paper applies to Indopacific coral reefs, not the Caribbean (acroporid recruitment between those two oceans differs significantly) and Suchley and Alvarez-Filip 2017 study was of adult coral growth and never once mentions settlement or recruitment.

Conclusions about what facilitates or inhibits coral recruitment could be a bit more circumspect. While it is correct that some coralline algae induce settlement and metamorphosis in corals, biofilms on smooth non-coralline surfaces do as well (see Fig. 2 in Arnold and Steneck 2011). After a ship grounding, biofilms would quickly become established. Also note that non-coralline crusts such as Peyssonnelia do not strongly inhibit coral settlement (Arnold and Steneck 2011). Nevertheless, undoubtedly macroalgal abundance would quickly dominate reef surfaces and effectively reduce all coral recruitment.

Rylaarsdam, K.W., 1983. Life histories and abundance patterns of colonial corals on Jamaican reefs. Marine ecology progress series. Oldendorf, 13(2), pp.249-260.

Experimental design

The design is fine. It is the conclusion that the abundance of macro algae is why reef building corals were rare to recruit

Validity of the findings

See #1 above. The main conclusions are not supported by their data or the literature.

They would be fine to conclude that the colonization of macro algae reduced coral recruitment - but the idea that the macro algae causes the absence of framework coral recruitment is unsupported.

Reviewer 2 ·

Basic reporting

The manuscript by Victoria-Salazar et al. presents interesting data on the condition of a Mexican reef system following an acute disturbance (a shipwreck). The authors measure multiple facets of the system to try to understand what feedback mechanisms may be responsible for the system’s supposed lack of recovery since the disturbance.

The manuscript is overall well-written and well-structured. However, there are several aspects I feel could be better elaborated upon or included to strengthen the manuscript. Also, some of the presented results are also unclear, preventing clear interpretations and conclusions to be drawn.

In terms of the background/context, I think the authors should include some more information on the sites’ earlier benthic composition (i.e. coral [genera] cover, algal cover) from previous studies. The authors refer to another study, but the reader shouldn’t need to then look separately at that study for this key piece of information from which comparisons are later drawn. For example, in line 88, the authors state their hypothesis that they ‘expected to find evidence for a relatively unsuccessful re-establishment of a healthy coral-dominated patch reef’, yet the prior 'healthy' composition has not been clarified. The reference given for the prior coral community is ‘Reefkeeper: Cancun reef sites dominated by algae’ but I can’t find this reference online and it seems incomplete in the reference list. It would also be good to draw some comparisons to other nearby reefs to place the results into context in terms of how the system compares to other areas. Such comparisons to other sites around the area and to earlier composition would be important to know if the community composition of the impacted site is a result of the system being locked in a degraded state since the ship grounding, or rather if this composition is reflective of reefs in the area (due to chronic stressors). Also, a clearer presentation of the significant differences between the two sectors would allow for more firm conclusions to be drawn in terms of the impact of the disturbance (i.e. what components of the community were determined to be significantly different between the two sectors).

Also, the authors mention several times that the area has been established as a marine protected area since the disturbance 16 years prior. However, while the authors refer to this protection and the importance of herbivorous fish several times (e.g. lines 42, 350-351), there is no fish data (only Diadema abundance is measured) presented to demonstrate whether the protected area is effectively protecting e.g. herbivorous fish biomass. Perhaps another study has measured this that can be cited?

Experimental design

In terms of the experimental design and methodology, I have a couple of suggestions for the authors.

When calculating coral cover, I would recommend the authors to better justify (or include a reference to support) their calculation of colony cover area. For a coral like P. astreoides (the dominant species found) I would think this could lead to a large underestimation of cover.

The authors measure densities of Diadema in quadrats which from my experience is a slightly unusual approach that is unlikely to yield reflective numbers (most studies measure Diadema in belt transects e.g. as per the AGGRA protocol). Justification of this choice of method should be stronger (i.e. if this is a common method in the area and generates comparable data to other studies).

The description of the collection of algal cover suggests that only algal cover was only quantified in 5 x 10cm2 areas were quantified per quadrat which seems a very small area from which to extract algal cover. Can the authors justify why this estimation was limited to such a small area?

Validity of the findings

I think it would be important to state clearly how this study complements or builds on the 2017 study from the same authors (Did the community structure of a coral reef patch affected by a ship grounding recover after 15 years? Merging historical and recent data sets) which also studies the coral and algal community, juvenile coral abundance and sea urchin abundance of the sites included in the current study up to an including 2013 (the first year of data collection for this study).

Some of the findings are also difficult to follow. For example, I am unclear on the results presented regarding algal categories. The authors state results such as ‘increases in non-CCA algae’ and ‘decreases in CCA’ over time. However, the supplementary figures do not show an increase in non-CCA over time – while fleshy algae is marginally higher at the third time point in S6, there is a lot of variation so it is unlikely significantly different and then the fourth time point has lower cover again. It also doesn’t look like CCA has decreased over the four time points from inspecting S6 or S7. It is generally unclear throughout the results and discussion section which differences that are presented have been determined to be significant. Importantly, for any models testing differences in algal groups over time/between sectors, the authors must ensure that quadrat is kept as a random effect based on the way the data was collected.

In terms of statistics, I comment the authors on undertaking some complex Bayesian statistical approaches. However, I feel the approach could be simplified and/or presented better to help the reader follow (i) what was tested and (ii) what relationships were actually significant. Similarly for the figures, I think it would be helpful to include a more understandable straightforward figure in the main manuscript that shows the differences via bar or boxplot in key groups between IS and NS and between years (i.e. a simplified version of S6/S7 with only key/dominant groups shown). Particularly, I find figure 3 very difficult to understand (other than there being higher occurrence probability of NFB coral recruiting), and some additional explanation would be key here.

Additional comments

Line 42: ‘available recruitment surface’ could be replaced by ‘available substrate for coral recruitment’

Lines 44-45: colonizable substrate leads to the successful settlement etc. of all coral species, not only morphologically complex types.

Line 53: I suggest to replace the older references here with Norström et al. 2009.

Norström, A. V., Nyström, M., Lokrantz, J., & Folke, C. (2009). Alternative states on coral reefs: beyond coral–macroalgal phase shifts. Marine ecology progress series, 376, 295-306.

Line 63: ‘spill’ should be ‘spills’

Line 65: recovery can also be driven by regrowth and natural fragmentation of remnant corals after such a disturbance, not only coral recruitment.

Lines 66-69: I am unsure how these two reported patterns are ‘complementary’ as one discusses the recovery of a coral-dominated system and the other is about opportunistic species preventing [coral recruitment?] on available surfaces following an acute disturbance. They seem to be rather opposing that complementary.

Lines 76-77: suggest changing to ‘Here, we study a coral reef patch that has been assigned as a no-take area since it was partially affected by a ship-grounding in 1997. We ask whether…’

Lines 80-81: unsure why this is defined as ‘occurrence probabilities’ rather than just ‘occurrence’ of small colonies, as this is based on simple counts of small corals <4cm.

Line 85: turf algae doesn’t necessarily impede coral recruitment – see Mumby et al. 2013 who find that coral recruitment can be high in turfs <5mm in height (see also recommendations by Flower et al. 2017). This may provide some support for why turf and fleshy algae occur on opposite sides of PC1 in Figure 2.

Mumby, P. J., Bejarano, S., Golbuu, Y., Steneck, R. S., Arnold, S. N., Van Woesik, R., & Friedlander, A. M. (2013). Empirical relationships among resilience indicators on Micronesian reefs. Coral Reefs, 32(1), 213-226.

Flower, J., Ortiz, J. C., Chollett, I., Abdullah, S., Castro-Sanguino, C., Hock, K., ... & Mumby, P. J. (2017). Interpreting coral reef monitoring data: A guide for improved management decisions. Ecological indicators, 72, 848-869.

Line 102: ‘stripe’ should be ‘strip’

Line 112: ‘which were dominated by opportunistic corals and fleshy algae with covers > 50%’ is difficult to interpret – what was the cover of opportunistic corals? Also, if fleshy algae already had cover >50% prior to the ship grounding then this would suggest the community has not been changed long-term by the ship grounding. This is why it is so important for earlier compositions to be presented in the manuscript.

Lines 140: Can the authors clarify better what is meant by signs of contraction?

Line 176: At this point the abbreviation NFB is not defined (nor the species included in this category).

Line 177: I assume ‘sector condition’ refers to IS or NS but this is not clear in the methods (only the caption of Fig. 4).

Lines 206-212: The differences identified between reefs and over time should be clarified better (in addition to the PCoA) – i.e. what differences were significant between the sectors and how exactly did the community change over the study period? Changes in algal composition over 6 months may be expected due to seasonal variation, so a clearer overview of this may help draw interpretations.

Line 215: It would be good to separate the juvenile corals by sectors (IS and NS) as it is unclear what was found where which is important given the comparison of the two areas.

Line 217: I think it is only necessary to state mean diameter ± SD at the first presentation, so it could be removed from here forward in this section.

Line 228: the algal groups are presented in many different ways throughout the manuscript text and the supplement (i.e. CCA can be presented as CCA/RP algae/recruitment promoting algae/algae promoting coral recruitment). I suggest to ensure consistency throughout to promote readability and interpretation.

Lines 267-269: fleshy algae is not a coral species (i.e. put the comma after P. astreoides to separate it from fleshy algae)

Line 273: As stated earlier in my comments, it is not clear that cover of non-CCA is increasing, and I am unsure why the authors conclude that non-CCA is gradually replacing the non-framework building corals (e.g. P. astreoides cover seems quite constant over time).

Lines 299-300: The authors state here that the impacted sector was previously dominated by framework-building corals – can it be shown somewhere in the text in the discussion or the introduction what the pre-disturbance cover of FB and NFB corals was?

Line 313: I think ‘apparently’ should be exchanged for ‘potentially’ for the three factors discussed as influencing coral recruitment.

Line 338: if there are standing dead Acropora skeletons, this suggests mortality from something other than the ship grounding. Could the authors elaborate on the causes of this mortality?

---

## Round 0.2 · accepted · Accept

The manuscript has now been re-evaluated by one of the previous reviewers and I agree with the assessment and am happy to accept the manuscript in its current form. I agree with the reviewer's assessment that removing some of the abbreviations in the manuscript could improve readability. However, I leave this decision to the authors and would like to thank you for your thorough revision.

PS: Consider changing "inhibition successional mechanism" to "inhibition of successional mechanisms" in the abstract during the proofing stage.

Reviewer 1 ·

Basic reporting

The manuscript as revised is very good. I think it would be easier to read if framework corals and non-framework corals were presented with those words rather than just their initials.

Relevant literature was cited.

Experimental design

It is fine as is.

Validity of the findings

Very good. In the response to my earlier review the authors stated: "that Peyssonnelids "as non-coralline, taxonomic and phylogenetic review of the group confirm that the genus includes coralline species distributed in the Caribbean. "

That is incorrect and the statement is not supported by the literature listed. Nor did those studies study coral recruitment.

This is not a serious problem for the manuscript but it does reflect some confusion among the authors. I simply wanted to point that out.

Additional comments

Paper as revised is fine.